

# Multiplex polymerase chain reaction (PCR) with Nanopore sequencing for sequence-based detection of four tilapia pathogens

Jérôme Delamare-Deboutteville[1,*], Watcharachai Meemetta[2,*], Khaettareeya Pimsannil[2], Han Ming Gan[3], Laura Khor[1], Mohan Chadag[1], Ha Thanh Dong[4] and Saengchan Senapin[2,5]

[1] Aquatic Food Biosciences, WorldFish, Batu Maung, Penang, Malaysia
[2] Fish Health Platform, Center of Excellence for Shrimp Molecular Biology and Biotechnology (Centex Shrimp), Faculty of Science, Mahidol University, Bangkok, Thailand
[3] Microbial Genomics, Patriot Biotech Sdn Bhd, Bandar Sunway, Selangor, Malaysia
[4] School of Environment, Resources and Development, Asian Institute of Technology, Pathum Thani, Thailand
[5] National Science and Technology Development Agency (NSTDA), National Center for Genetic Engineering and Biotechnology (BIOTEC), Pathum Thani, Thailand
* These authors contributed equally to this work.

Corresponding authors
Jérôme Delamare-Deboutteville, J.Delamare@cgiar.org
Saengchan Senapin, saengchan@biotec.or.th, senapin2010@gmail.com

## ABSTRACT

**Background**. Tilapia aquaculture faces significant threats posed by four prominent pathogens: tilapia lake virus (TiLV), infectious spleen and kidney necrosis virus (ISKNV), *Francisella orientalis*, and *Streptococcus agalactiae*. Currently, employed molecular diagnostic methods for these pathogens rely on multiple singleplex polymerase chain reactions (PCR), which are time-consuming and expensive.
**Methods**. In this study, we present an approach utilizing a multiplex PCR (mPCR) assay, coupled with rapid Nanopore sequencing, enabling the one-tube simultaneous detection and one-reaction Nanopore sequencing-based validation of four pathogens.
**Results**. Our one-tube multiplex assay exhibits a detection limit of 1,000 copies per reaction for TiLV, ISKNV, and *S. agalactiae*, while for *F. orientalis*, the detection limit is 10,000 copies per reaction. This sensitivity is sufficient for diagnosing infections and co-infections in clinical samples from sick fish, enabling rapid confirmation of the presence of pathogens. Integrating multiplex PCR and Nanopore sequencing provides an alternative approach platform for fast and precise diagnostics of major tilapia pathogens in clinically sick animals, adding to the available toolbox for disease diagnostics.

# INTRODUCTION

Tilapia (*Oreochromis* spp.) is one of the most widely farmed freshwater fish species globally due to its high adaptability, fast growth, and excellent meat quality. Global production is

estimated at 6,100,719 tonnes in 2020 (*FAO, 2024*). By providing sustenance, employment opportunities, and domestic and export revenues, this valuable species supports large populations worldwide (*Wang & Lu, 2016*). In the past decade, tilapia production has almost doubled (*FAO, 2024*), attributed mainly to its ease of cultivation, market demand, and stable pricing (*Wang & Lu, 2016*).

However, the production of tilapia has been threatened by several viral and bacterial diseases that can cause significant economic losses (*Debnath et al., 2023*; *Soto et al., 2025*; *Senapin et al., 2025*). Among the most important tilapia pathogens are tilapia lake virus (TiLV), infectious spleen and kidney necrosis virus (ISKNV), *Francisella orientalis* (FNO), and *Streptococcus agalactiae* (SAG) also called group B *streptococcus* (GBS) (*Kawasaki et al., 2018*; *Machimbirike et al., 2019*; *Mabrok et al., 2021*; *Haenen et al., 2023*; *Alathari et al., 2023*). Co-infections with these and other pathogens can exacerbate health and production issues in tilapia (*Xu & Shoemaker, 2025*). For example, from a 2015 to 2016 longitudinal study, SAG was the main bacterium detected in Nile tilapia farmed in a Brazilian reservoir and was mostly found concurrently with one or two bacteria, such as FNO and *Aeromonas hydrophila* (*Delphino et al., 2019*). Similarly, ISKNV has been detected in co-infections with SAG and other bacteria during high mortality events in Thailand (*Dong et al., 2015*) and Ghana (*Ramírez-Paredes et al., 2021*). Concurrent infections of TiLV and *Aeromonas* spp. have also been reported in mortality outbreaks in Egypt (*Nicholson et al., 2017*) and Thailand (*Nicholson et al., 2020*). Notably, mass mortality events of red hybrid tilapia have confirmed the presence of TiLV, SAG, and *A. hydrophila*, with mortality rates reaching up to 70% (*Basri et al., 2020*). Effective diagnosis and monitoring of these pathogens are crucial for disease management and control (*Dong et al., 2023*). Traditional diagnostic techniques such as virus and bacterial isolation, histopathology, immunofluorescence assay (IFA), and enzyme-linked immunosorbent assay (ELISA) are time-consuming, labor-intensive, and require specialized equipment and expertise (*Dong et al., 2023*). In contrast, molecular methods such as polymerase chain reaction (PCR) and quantitative PCR (qPCR) have gained widespread use in pathogen detection due to their high sensitivity and specificity, and rapid turnaround times (*Soto et al., 2010*; *Kralik & Ricchi, 2017*; *Liamnimitr et al., 2018*; *Waiyamitra et al., 2018*; *López-Porras et al., 2019*; *Ramírez-Paredes et al., 2021*; *Taengphu et al., 2022*; *Dong et al., 2023*).

During disease outbreaks, affected fish often harbor multiple pathogens (*Dong et al., 2015*; *Delphino et al., 2019*; *Abdel-Latif et al., 2020*; *Basri et al., 2020*; *Ramírez-Paredes et al., 2021*). Traditional diagnostic methods such as PCR and qPCR typically require separate tests for each pathogen, making the process time-intensive and costly, particularly when identifying multiple pathogens. In contrast, multiplex PCR allows for the simultaneous detection of multiple pathogens in a single sample by using multiple sets of primers, each specifically designed to bind to a specific DNA target. Although the sensitivity of multiplex PCR is generally lower than that of single-pathogen PCR or qPCR (*Elnifro et al., 2000*), it remains a practical and efficient diagnostic method, especially for clinical samples where pathogen loads are often high. Although multiplex PCR is efficient, the use of multiple primer sets in a single reaction can occasionally result in non-specific amplification. These challenges can be mitigated through careful primer design and optimization (*Henegariu*

*et al., 1997*), underscoring the importance of sequence-based verification for accurate pathogen identification.

Sanger sequencing, a widely used DNA sequencing method, has limitations that restrict its use in detecting pathogens effectively in complex biological samples. It is relatively low throughput, sequencing one DNA fragment at a time, and requires a relatively high amount of pure DNA fragments (1 μg). Moreover, prior knowledge of the target region is essential for primer design, making it unsuitable for identifying novel or unexpected pathogens. Its reliance on specialized laboratory equipment and infrastructure further limits its portability, rendering it impractical for field-based diagnostics or use in resource-limited settings.

In contrast, Oxford Nanopore Technology (ONT) addresses these limitations by enabling on-site amplicon sequencing and offering several advantages for accurate pathogen detection (*Delamare-Deboutteville et al., 2021*). ONT requires only 50 ng of DNA per sample and can simultaneously sequence multiple amplicons of varying sizes, particularly short fragments (~200 bp). This capability provides high sensitivity and precision for strain-level identification and differentiation—critical for understanding pathogen diversity, epidemiology, and tailoring disease management strategies. Its portability, scalability, and real-time sequencing capabilities make it especially suitable for field-based diagnostics and resource-limited settings, allowing rapid and reliable pathogen identification while avoiding delays associated with sample transport to distant centralized laboratories. By confirming target identity and resolving closely related strains, species, or novel variants, Nanopore sequencing complements PCR, offering insights beyond what PCR alone can achieve. Consequently, integrating multiplex PCR with ONT-based genotyping represents a powerful approach for aquaculture, enabling precise pathogen detection and characterization to enhance biosecurity and disease management.

We present a multiplex PCR assay with a detection level suitable for simultaneously identifying TiLV, ISKNV, *F. orientalis*, and *S. agalactiae* in sick tilapia samples. Using the portable Nanopore sequencing platform, we confirmed the presence of these pathogens at the sequence level and gathered genetic information from the amplicons. This assay offers a rapid and practical solution for detecting and genetically characterizing these pathogens, with the potential to complement existing diagnostic tools and inform targeted surveillance and control strategies.

## MATERIALS & METHODS

### Re-used text from preprint

Portions of this text were previously published as part of a preprint (https://www.biorxiv.org/content/10.1101/2023.05.13.540096v3).

### Ethics declarations

The authors confirm that they that adhered to the journal's ethical policies, as noted on the journal's author guidelines page. No ethical approval was required as no animals were used in this study. Virus sequences were generated from archived samples.

## Clinical samples and nucleic acid extraction

We utilized archival clinical samples of fry, fingerling, juvenile and adult Nile tilapia, red tilapia, and Asian sea bass from challenge experiments or from various disease outbreaks between 2015 and 2020. Our investigation included samples that were either confirmed to be caused by a single pathogen using PCR diagnosis or suspected to have resulted from co-infections. All sample details are comprehensively listed in Table 1. To extract the nucleic acid from the samples, some were processed using the PathoGen-spin DNA/RNA extraction kit from iNtRON Biotechnology, while others were archival RNA samples extracted using Trizol reagent from Invitrogen, and DNA samples extracted using the conventional phenol/chloroform ethanol precipitation method as described by *Meemetta et al. (2020)*. If done close to the farm, the diagnostic workflow from the point of sample collection to the final data analysis can take less than 12 h. The processes include nucleic acid extraction, multiplex PCR, library preparation, Nanopore sequencing, and data analysis, as illustrated in Fig. 1.

## Primers used in this study

We obtained the primers for the four selected target pathogens from Bio Basic (Markham, Ontario, Canada), and the primer sequences are summarized in Table S1. Primers for TiLV, ISKNV, *F. orientalis*, and *S. agalactiae* were reported in previous studies (*Yang et al., 2013*; *Dong et al., 2016*; *Paria et al., 2016*; *Leigh et al., 2018*; *Kawato et al., 2021*; *Taengphu et al., 2022*). The expected amplicon sizes were 137, 190, 203, and 351 bp, respectively. The specificity of each primer pair was assessed *in silico* using the Primer-BLAST program (https://www.ncbi.nlm.nih.gov/tools/primer-blast/).

## Plasmid positive controls and the analytical sensitivity assay

All plasmid-positive controls except for *S. agalactiae* (constructed in this study) were obtained from our previous studies (Table S2). Positive plasmid control containing *S. agalactiae groEL* partial gene was obtained by cloning a 351 bp-*groEL* amplified fragment purified before being ligated into pGEM-T easy vector (Promega). A recombinant plasmid was subjected to DNA sequencing (Macrogen). The copy number of each plasmid was calculated based on its size in base pair (bp) and amount in nanogram (ng) using a web tool at http://sciprim.com/html/copyNumb.v2.0.html. A combination of four recombinant plasmids was mixed for the multiplex PCR sensitivity assay. This mixture was then subjected to 10-fold serial dilution, resulting in a 1 to $10^6$ copies/$\mu$l range. Subsequently, 4 $\mu$l of the diluted series was used in the multiplex PCR reaction. To simulate a clinical sample from a fish, each PCR reaction was spiked with 50 ng of RNA extracted from a healthy tilapia.

## Multiplex PCR condition optimization and the detection of clinical samples

Previous studies claimed that qPCR reagents usually contain PCR additives and enhancers that can improve amplification efficiency (*Karunanathie et al., 2022*). To optimize our multiplex PCR, we used KAPA SYBR FAST One-Step qRT-PCR master mix (Roche), known to contain such additives. Each reaction was prepared in a 25 $\mu$l volume, comprising 1X master mix, four $\mu$l of template, and 80–240 nM of each primer pair (Table S3).

**Table 1  Sources of clinical tilapia samples and PCR detection results.**

| Code | Sample_name | Life stage | Country | Year isolation | Previous diagnosis sPCR | mPCR | Refs/comments |
|------|-------------|-----------|---------|----------------|-------------------------|------|---------------|
| 1 | Healthy NT 1 | juvenile (NT) | Thailand | 2020 | – | – | *Taengphu et al. (2022)* |
| 2 | Healthy NT 2 | juvenile (NT) | Thailand | 2020 | – | – | *Taengphu et al. (2022)* |
| 3 | *r2 _NT F1*# | fingerling (NT) | Thailand | 2020 | TiLV (++) | TiLV* | *Taengphu et al. (2022)* |
| 4 | *r2 _NT F2*# | fingerling (NT) | Thailand | 2020 | TiLV (++) | TiLV* | *Taengphu et al. (2022)* |
| 5 | *r2 _NT F3*# | fingerling (NT) | Thailand | 2020 | TiLV (++) | TiLV* | *Taengphu et al. (2022)* |
| 6 | *r2_NT F4*# | fingerling (NT) | Thailand | 2020 | TiLV (++) | TiLV* | *Taengphu et al. (2022)* |
| 7 | *r2_NT F5*# | fingerling (NT) | Thailand | 2020 | TiLV (++) | TiLV* | *Taengphu et al. (2022)* |
| 8 | NT fingerling pond C1 | fingerling (NT) | Thailand | | TiLV (+) | – | *Taengphu et al. (2022)* |
| 9 | NT fingerling pond C2 | fingerling (NT) | Thailand | | TiLV (+) | – | *Taengphu et al. (2022)* |
| 10 | NT female brood | adult (NT) | Thailand | 2020 | TiLV (+) | – | *Taengphu et al. (2022)* |
| 11 | NT male brood | adult (NT) | Thailand | | TiLV (+) | – | *Taengphu et al. (2022)* |
| 12 | NT juvenile 4 | juvenile (NT) | Thailand | | TiLV (+) | – | *Taengphu et al. (2022)* |
| 13 | RT F1 | fingerling (RT) | Thailand | | FnO (+) | – | *Nguyen et al. (2016)* |
| 14 | RT F2 | fingerling (RT) | Thailand | 2015 | FnO (+) | – | *Nguyen et al. (2016)* |
| 15 | RT F3 | fingerling (RT) | Thailand | | FnO (+) | – | *Nguyen et al. (2016)* |
| 16 | RT F4 | fingerling (RT) | Thailand | | FnO (+) | – | *Nguyen et al. (2016)* |
| 17 | NT 1.2 | fingerling (NT) | Thailand | | TiLV (+) | – | *Dong et al. (2017)* |
| 18 | NT 1.3 | fingerling (NT) | Thailand | | TiLV (+) | – | *Dong et al. (2017)* |
| 19 | NT 2.1 | fingerling (NT) | Thailand | 2015 | TiLV (+) | – | *Dong et al. (2017)* |
| 20 | NT 2.3 | fingerling (NT) | Thailand | | TiLV (+) | – | *Dong et al. (2017)* |
| 21 | NT 3.1 | fingerling (NT) | Thailand | | TiLV (+) | – | *Dong et al. (2017)* |
| 22 | NT 3.3 | fingerling (NT) | Thailand | | TiLV (+) | – | *Dong et al. (2017)* |
| 23 | NT Gh 21 | juvenile (NT) | West Africa | | ISKNV (++) | ISKNV* | - |
| 24 | NT Gh 22 | juvenile (NT) | West Africa | | ISKNV (++) | ISKNV* | - |
| 25 | *r2 _NT Gh 23*# | juvenile (NT) | West Africa | 2019 | ISKNV (++) | ISKNV* | – |
| 26 | *r2 _NT Gh 24*# | adult (NT) | West Africa | | ISKNV (++) | ISKNV* | – |
| 27 | *r2 _NT Gh 25*# | juvenile (NT) | West Africa | | ISKNV (++) | ISKNV* | – |
| 28 | RT EX 01 | fry (RT) | Thailand | | FnO (++) | FnO## | *Nguyen et al. (2019)* |
| 29 | RT EX02 | fry (RT) | Thailand | | FnO (++) | FnO## | *Nguyen et al. (2019)* |
| 30 | *r2_RT EX03*# | fry (RT) | Thailand | 2019 | FnO (++) | FnO## | *Nguyen et al. (2019)* |
| 31 | *r2_RT FM1a*# | adult (RT) | Thailand | | FnO (++) | FnO## | *Nguyen et al. (2019)* |
| 32 | *r2_RT M1*# | adult (RT) | Thailand | | FnO (++) | FnO## | *Nguyen et al. (2019)* |
| 33 | *r2_RT FM1b*# | adult (RT) | Thailand | | FnO (++) | FnO## | *Nguyen et al. (2019)* |
| 34 | NT PV F8 | adult (NT) | Thailand | | Not done | * | – |
| 35 | NT PV F9 | adult (NT) | Thailand | 2020 | Not done | ## | – |
| 36 | NT PV F10 | adult (NT) | Thailand | | Not done | ## | – |
| 37 | *r3 _NT Group B*# | fry (NT) | Thailand | | SAG (++) | SAG | – |
| 38 | *r3 _NT Zone A*# | fry (NT) | Thailand | 2019 | SAG (++) | SAG* | – |
| 39 | *r3 _NT Zone B*# | fry (NT) | Thailand | | SAG (++) | SAG | – |

| Code | Sample_name | Life stage | Country | Year isolation | Previous diagnosis sPCR | mPCR | Refs/comments |
|------|-------------|------------|---------|----------------|-------------------------|------|---------------|
| 40 | *r1_TILV_1*# | fingerling (RT) | Thailand | 2018-07 | TiLV | TiLV | *Thawornwattana et al. (2021)* |
| 41 | *r1_ISKNV_1*# | fry (AS) | Thailand | 2018-11 | ISKNV | ISKNV | *Kerddee et al. (2021)* |
| 42 | *r1_FNO_1*# | fingerling (RT) | Thailand | 2015 | FnO | FnO | *Nguyen et al. (2016)* |
| 43 | *r1_SAG_1*# | fry (NT) | Thailand | 2019-07 | SAG | SAG | |
| 44 | *r3_TILV50*# | fingerling (RT) | Thailand | 2019-07 | TiLV | TiLV | TiLV PCR amplicon of 100 ng |
| 45 | *r3_TILV30*# | fingerling (RT) | Thailand | 2019-07 | TiLV | TiLV | TiLV PCR amplicon of 75 ng |
| 46 | *r3_TILV15*# | fingerling (RT) | Thailand | 2019-07 | TiLV | TiLV | TiLV PCR amplicon of 50 ng |
| 47 | *r3_TILV05*# | fingerling (RT) | Thailand | 2019-07 | TiLV | TiLV | TiLV PCR amplicon of 25 ng |
| 48 | *r3_SAG50*# | fry (NT) | Thailand | 2019-07 | SAG | SAG | SAG PCR amplicon of 100 ng |
| 49 | *r3_SAG30*# | fry (NT) | Thailand | 2019-07 | SAG | SAG | SAG PCR amplicon of 75 ng |
| 50 | *r3_SAG15*# | fry (NT) | Thailand | 2019-07 | SAG | SAG | SAG PCR amplicon of 50 ng |
| 51 | *r3_SAG05*# | fry (NT) | Thailand | 2019-07 | SAG | SAG | SAG PCR amplicon of 25 ng |
| 52 | *r1_4PAT_1*# | N/A | Thailand | N/A | 4PAT | 4PAT | PCR amplicons 4 PAT (combined1) |
| 53 | *r1_4PAT_DIL_1*# | N/A | Thailand | N/A | 4PAT | 4PAT | PCR amplicons 4 PAT (combined2) |
| 54 | *r3_4PAT_DIL_3*# | N/A | Thailand | N/A | 4PAT | 4PAT | PCR amplicons 4 PAT (combined3) |

**Notes.**

Notes and abbreviations: sPCR, singleplex PCR; mPCR, multiplex PCR; NT, Nile tilapia; RT, red tilapia; AS, Asian sea bass; −, negative test; ++, high pathogen load; +, low pathogen load.

\*suspected dual infections with other pathogen(s).

##probably non-specific products.

TiLV, tilapia lake virus; ISKNV, infectious spleen and kidney necrosis virus; FnO, *Francisella noatunensis* subsp. *orientalis*; SAG, *Streptococcus agalactiae*; 4PAT, mixture of PCR amplicons 4 pathogens (TiLV, ISKNV, FnO, SAG).

Samples sequenced are italicized with#, and the designation "r1" to "r3" indicates the Nanopore run number.

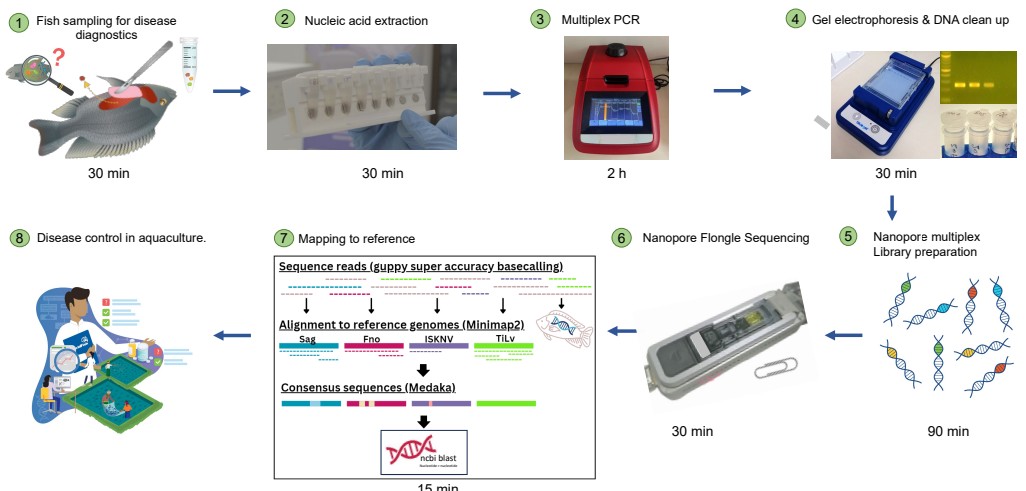

**Figure 1** **Simplified workflow illustration of diseased fish sampling on farm to Nanopore sequencing results.** The entire process can take less than 12 h.

We determined the optimal annealing temperature (Ta) using gradient PCR with Ta ranging between 55−65 °C. To identify the best combination of PCR components, we added ammonium sulfate, BSA, dNTPs, and MgCl2 in different proportions. The final cycling conditions comprised a reverse transcription step at 42 °C for 5 min, followed by inactivation at 95 °C for 3 min, and 40 cycles of 95 °C for 10 s and 60 °C for 30 s, with a final extension step at 60 °C for 5 min (Table S3). We then analyzed 10 µl of each product by electrophoresis on a 3.5% agarose gel stained with ethidium bromide. The newly optimized multiplex PCR assay was subsequently used to detect the presence of TiLV, ISKNV, *F. orientalis*, and *S. agalactiae* in archival clinical samples (Table 1). Most of these samples were previously tested for a single pathogen using PCR or qPCR assays.

## Quantification of pathogens by qPCR assays

A total of 15 clinical samples, whose mPCR products underwent Nanopore sequencing, were analyzed to determine the presence and quantity of each pathogen. The established protocols (*Leigh et al., 2018*; *Kawato et al., 2021*; *Taengphu et al., 2022*) were followed to detect TiLV, ISKNV, and SAG, as described in Table S4. For FnO qPCR, the same primers used in mPCR were used in this study's SYBR Green-based qPCR assay.

## Nanopore sequencing

The present study employed amplicons generated from both single and multiplex PCR reactions (Table 1, samples highlighted in grey) as templates for library preparation using the ligation sequencing kit (SQK-LSK109) and the native barcoding expansion 1–12 kit (EXP-NBD104) according to the standard protocols of ONT adapted for the Flongle flow cell. Three sequencing runs (r1-r3) were conducted, with 250 ng of PCR product per sample and a unique native barcode (BC) assigned to each sample. Run 1 (r1) involved sequencing amplicons obtained from individual PCR reactions and a combination of these amplicons (Table 1). Run 2 (r2) focused on sequencing multiplex PCR (mPCR) products derived from clinical samples. Run 3 (r3) employed mPCR products from the remaining clinical samples. Additionally, different concentrations of single PCR products were included in this run. The library of pooled barcoded samples was subjected to a Short Fragment Buffer (SFB) wash before the final elution step of the protocol. The DNA concentration was quantified at every step using the Qubit assay. Subsequently, the prepared library was loaded onto a Flongle flow cell (FLO-FLG106), following the Nanopore standard protocol, and each Flongle flow cell was fitted to a Flongle adapter (FLGIntSP) for MinION.

## Reads filtering and read abundance calculation

Basecalled reads in fastq format were primer-trimmed with cutadapt v4.3 (*Martin, 2011*), and reads that have been trimmed with length ranging from 75–400 bp will be retained for the subsequent analysis, leaving out overly short or long reads without intact primer sequence on both ends. The filtered reads were aligned to the four pathogen gene segments using Minimap2 v2.17 (*Li, 2018*). Reads that aligned uniquely (only a single hit reported) with more than 80% coverage to the target region were used for abundance calculation and variant calling (consensus generation). Raw, trimmed, and aligned read statistics were calculated using SeqKit v2.2 (*Shen et al., 2016*). Reads failing to align at this stringent

level were extracted and re-aligned using the default Minimap2 setting, followed by a less stringent blastN v2.15.0 default alignment. The host genome was included in the reference sequence to assess the fraction of reads mapping to the host.

## Generation of consensus and variant sequences

Only reads with unique alignments were used as the template for Minimap2 v.2.17 and medaka v1.12.1 variant calling based on the ARTIC pipeline "Core Pipeline - artic pipeline"; "sars-cov-2-ont-artic-variant- calling/COVID-19-ARTIC-ONT". Briefly, within the pipeline, the reads were mapped to the reference gene segments of four pathogens, followed by variant calling using the medaka variant model r941_min_g507. Variant filtering was then performed based on mapping quality and read depth. The number of ambiguous bases in the consensus sequences was calculated using QUAST v5 (*Gurevich et al., 2013*). Only sequences without any ambiguous base were selected for subsequent variant sequence generation. Variants were generated using cd-hit v.4.8.1 (*Huang et al., 2010*), whereby sequences with 100% nucleotide identity and exact sequence length were clustered into the same genotype. The variant sequences were compared against the NCBI Blast nt database (accessed on 25 April 2023) to identify the top 5 BLAST hits for each sequence.

# RESULTS

## Primer testing by single PCR

Single PCR reactions were performed using each primer pair with its respective target template to validate the selected primer pairs for each pathogen. The amplification of the target regions was confirmed by amplicons of the expected sizes, as illustrated in Fig. S1. To simulate the expected products from multiplex PCR (mPCR), an equal amount of each individual PCR product was loaded into lane C of the Fig. S1. Gel electrophoresis analysis revealed distinct and separable bands, demonstrating the potential for further utilization in the multiplex PCR analysis. Note that the products generated from this step were subsequently employed in Nanopore sequence run 1 (r1).

## Multiplex PCR detection of four pathogens among clinical samples

We have successfully optimized a multiplex PCR assay to simultaneously detect four important tilapia pathogens, TiLV, ISKNV, *F. orientalis*, and *S. agalactiae*, in a single reaction. The detection sensitivity of the new assay in the presence of spiked host RNA was $10^3$ copies/reaction for TiLV, ISKNV, and *S. agalactiae*. In the presence of $10^4$ copies of each template, the assay could detect *F. orientalis* and three other pathogens (Fig. S2). The assay could detect each pathogen efficiently in tilapia clinical samples, as confirmed by distinct PCR bands (Fig. 2). Some suspected co-infections detected by PCR were confirmed using Nanopore sequencing. Among the 15 clinical samples analyzed, nine exhibited co-infections involving three to four pathogens (Fig. 3). The assay was able to detect TiLV in heavily infected samples (3–7) both on the gel and by Nanopore sequencing but failed to detect it on the gel for samples (8–12) with low levels of TiLV infection. Similarly, the highest number of reads for ISKNV were found in sample #25 (twice more

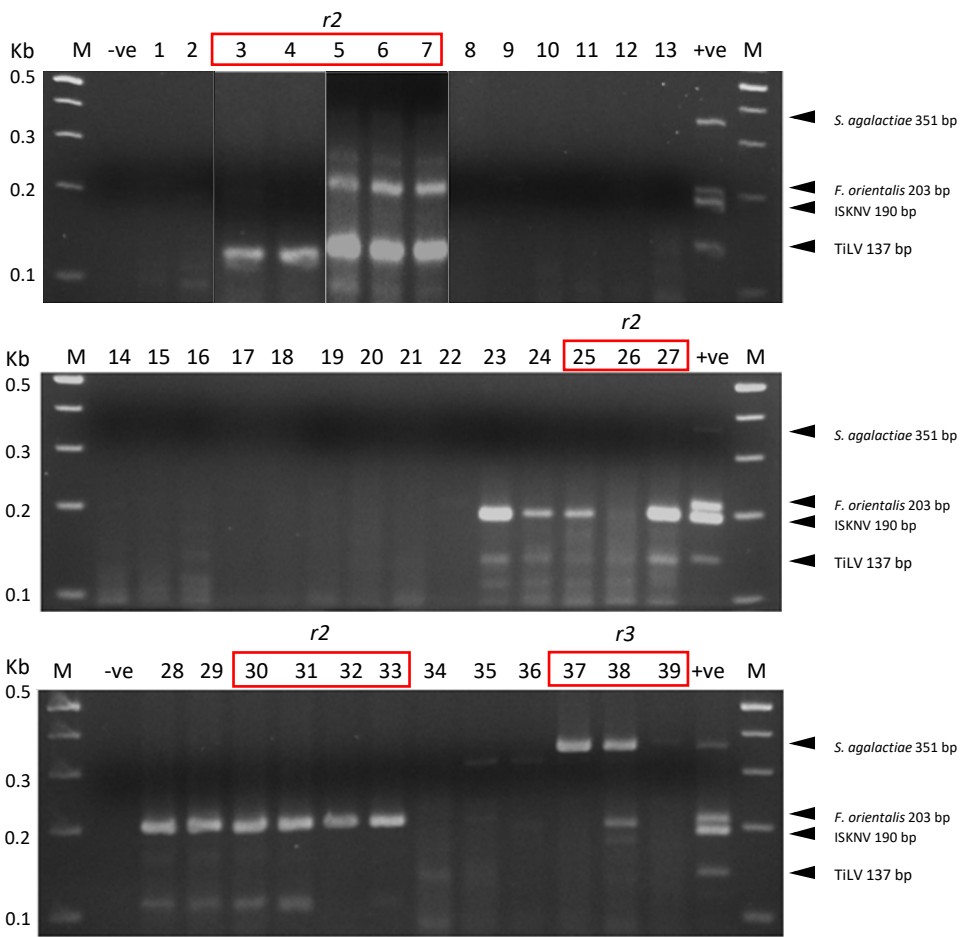

**Figure 2 Multiplex PCR detection of TiLV, ISKNV, *F. orientalis,* and *S. agalactiae* from clinically diseased tilapia.** Sample codes are listed in Table 1. The expected amplicon size for each pathogen is indicated on the right. -ve, no template control, +ve, mix positive control plasmids; M, 100 bp DNA marker (New England BioLabs). Samples marked *r2* to *r3* were subjected to Nanopore sequencing run numbers 2 and 3, respectively.

than in sample #27), yet it had a much fainter band on the gel (Table 1). There is a good correlation between read numbers and bands intensity for *F. orientalis* in four samples (# 30–33) but only a weak band for *S. agalactiae* in one sample (#39) compared to number of amplicons (Figs. 2 and 3, Table S5). In some samples, we detected dual infections with more than one pathogen, as evidenced by the presence of multiple bands, including a possible co-infection with *F. orientalis* and TiLV in samples 3–7. However, differentiation of *F. orientalis* and ISKNV was challenging as their respective bands have similar sizes (Fig. 2).

## Successful recovery of pathogen sequence variants from infected samples

A total of 246,756 reads were successfully demultiplexed from three separate Flongle runs (r1–r3), each containing different sample types (Table 1, Table S5). Following primer and

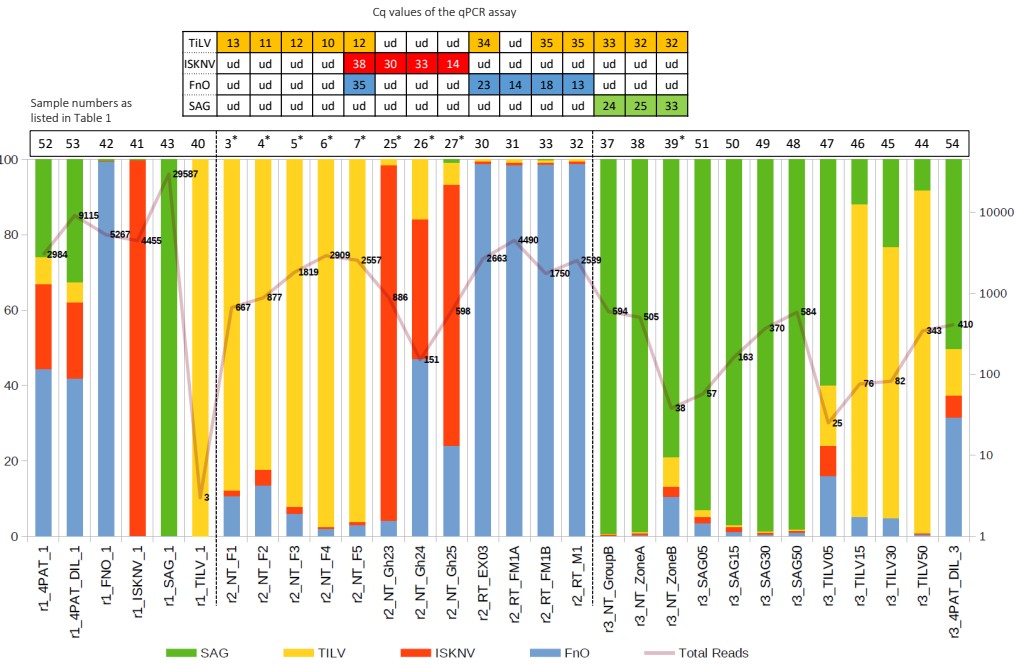

**Figure 3** **Percentage of reads mapping to each pathogen after primer trimming (left axis) and the total filtered read count for each sample (right axis).** Table at the top: single pathogen tested by qPCR assay, with Cq value indicated; ud, undetectable. The curve with numbers across the histogram represents the number of reads that are aligned to the reference sequences of the four pathogens. An asterisk (*) indicates, suspected dual infections with other pathogens.

length filtering, a substantial reduction in read count was observed, resulting in an average of 64% of reads being removed (range: 46%–87%) (Table S5). An additional 10% of reads were removed after alignment filtering. After these filtering steps, 75,203 reads remained, providing an average of 2,500 filtered reads per sample for subsequent read abundance and consensus sequence generation (Table S5).

Given the small sampling size and possibly highly conserved nature of some of the pathogen gene segments, one sequence variant was generated per pathogen gene segment except for the TiLV gene segment, whereby two sequence variants were recovered (Table 2). All recovered sequence variants exhibited 100% nucleotide identity to at least one publicly available sequence in the NCBI database. In addition, samples derived from the single PCR product inputs and their combined mixture were clustered in their respective sequence variants. In the case of the TiLV amplicons, we observed two unique sequence variants (Var 0 and Var 1) for TiLV gene segment 9, one of which displayed 100% nucleotide identity to the gene region of a relatively divergent TiLV strain from Vietnam (Table 2). This sequence variant was detected in samples r2_NT_F1 to r2_NT_F5, collected from the same sampling site simultaneously (Table 2).

**Table 2  List of variants for the pathogen gene region, along with the inferred variants and their top 5 NCBI BLAST hits.**

| Pat | Samples | Var | Top 5 BLAST hits | %ID | Accession |
|---|---|---|---|---|---|
| FnO | r1_4PAT_1,r1_4PAT_DIL_1, r1_FNO_1,r1_SAG_1 r2_NT_F1, r2_NT_F2,r2_NT_F3,r2_NT_F4, r2_NT_F5,r2_NT_Gh23, r2_NT_Gh24, r2_NT_Gh25,r2_RT_EX03, r2_RT_FM1A,r2_RT_FM1B, r2_RT_M1 r3_4PAT_DIL_3 | 0 | *Francisella noatunensis* subsp. *orientalis* strain LPM2-AR2019 | 100 | MN385384.1 |
| | | | *Francisella noatunensis* subsp. *orientalis* strain FO371 | 100 | CP022953.1 |
| | | | *Francisella noatunensis* subsp. *orientalis* strain FNO364 | 100 | CP022952.1 |
| | | | *Francisella noatunensis* subsp. *orientalis* strain FNO222 | 100 | CP022951.1 |
| | | | *Francisella noatunensis* subsp. *orientalis* strain FNO215 | 100 | CP022950.1 |
| ISKNV | r1_4PAT_1,r1_4PAT_DIL_1, r1_ISKNV_1 r2_NT_F2,r2_NT_Gh23, r2_NT_Gh24,r2_NT_Gh25 | 0 | ISKNV isolate ISKNV/10 | 100 | MT178422.1 |
| | | | ISKNV isolate ISKNV/48 | 100 | MT178418.1 |
| | | | Angelfish iridovirus AFIV-16 | 100 | MK689685.1 |
| | | | ISKNV isolate M6 | 100 | MK084827.1 |
| | | | ISKNV isolate SB04 | 100 | KY440040.1 |
| SAG | r1_SAG_1,r1_4PAT_1, r1_4PAT_DIL_1, r1_SAG_1 r3_4PAT_DIL_3,r3_NT_GroupB, r3_NT_ZoneA, r3_NT_ZoneB,r3_SAG05, r3_SAG15,r3_SAG30,r3_SAG50 | 0 | *Streptococcus agalactiae* strain 01173 | 100 | CP053027.1 |
| | | | *Streptococcus agalactiae* strain Sag153 | 100 | CP036376.1 |
| | | | *Streptococcus agalactiae* strain ZQ0910 | 100 | CP049938.1 |
| | | | *Streptococcus agalactiae* strain NJ1606 | 100 | CP026084.1 |
| | | | *Streptococcus agalactiae* strain BSE009 | 100 | CP020387.1 |
| TiLV | r1_4PAT_1,r1_4PAT_DIL_1 r3_4PAT_DIL_3, r3_TILV15, r3_TILV30, r3_TILV50 | 0 | TiLV isolate WVL19054 segment 9 | 100 | MN193531.1 |
| | | | TiLV isolate WVL19031-01A segment 9 | 100 | MN193521.1 |
| | | | TiLV isolate EC-2012 segment 9 | 100 | MK392380.1 |
| | | | TiLV isolate Til-4-2011 segment 9 | 100 | KU751822.1 |
| | | | TiLV AD-2016 Contig 20 | 100 | KU552140.1 |
| | | | TiLV isolate RIA2-VN-2019 segment 9 | 100 | ON376590.1 |
| | r2_NT_F1,r2_NT_F2,r2_NT_F3, r2_NT_F4,r2_NT_F5 | 1 | TiLV isolate WVL19054 segment 9 | 98.91 | MN193531.1 |
| | | | TiLV isolate WVL19031-01A segment 9 | 98.91 | MN193521.1 |
| | | | TiLV strain EC-2012 segment 9 | 98.91 | MK392380.1 |
| | | | TiLV isolate Til-4-2011 segment 9 | 98.91 | KU751822.1 |

**Notes.**

Pat, Pathogen; Var, Variants; %ID, % nucleotide identity.

The samples were grouped based on the sequencing run from which they were generated.

Abbreviations: FnO, *Francisella noatunensis* subsp. *orientalis*; ISKNV, infectious spleen and kidney necrosis virus; SAG, *Streptococcus agalactiae*;; TiLV, tilapia lake virus.

### Effect of template input on data distribution and pathogen detection specificity on the Nanopore platform

In the first run (r1), using well-defined pathogen nucleic acids as templates for the detection assay, confident alignments were observed for each sample to their expected pathogen gene segments, except for sample r1_TILV_1 which consisted purely of TiLV amplicons and did not sequence well enough to enable subsequent variant calling (Fig. 3 and Table 2). For sample 4PAT, which consisted of a combination of products from 4 pathogens, the read distribution was broadly similar, as was the case for sample 4PAT_DIL, which used a ten-fold diluted version of the template (Fig. 3). In the second run (r2), actual clinical samples with PCR-verified infections were used, and over 50% and up to 99% of reads were primarily mapped to the primary suspected pathogen gene fragments, with the remaining reads mapping to the other three pathogens. Interestingly, in the third run (r3), a significant percentage (>5%) of reads aligned to other pathogens were observed among samples spiked with different amounts of single amplified products (Fig. 3). For example, among the TiLV samples (TILV05, 15, 30, 50), the lowest amount of PCR product template (TILV05) resulted in a low number of TiLV reads and a higher percentage of reads belonging to non-TiLV pathogens. A similar trend was observed for pure SAG samples, with the most diluted SAG sample (SAG05) having the highest percentage of non-SAG reads. However, in contrast to the pure synthetic TiLV samples, the percentage of non-specific reads decreased to less than 2% in the SAG50 sample with the highest read count, while it remained around 10% in the TILV50 sample.

### Reads that failed stringent alignment to the pathogen gene panel were host-derived or partial pathogen sequences with lower nucleotide identity

Upon investigation of reads that failed to align, a significant proportion was found to map not only to the four pathogen gene segments but also to the host genome when a more lenient alignment configuration was employed (Fig. 4). Samples from run 2 (r2), predominantly derived from fish tissues, exhibited a higher percentage of reads mapping to the host genome than other samples. As expected, given the use of input PCR products as the template for all samples in run 1 (r1), no reads were found to align to the host genome when they were aligned with Minimap2 (default setting). On the contrary, a small portion of reads belonging to the host genome was found among different concentrations of single PCR products from run 3 (r3) (Fig. 4 and Table S5), which also consists of samples derived from tilapia organs (Table 1).

### Comparative advantages of mPCR + Nanopore over qPCR

The mPCR + Nanopore approach offers several advantages over qPCR, including greater efficiency in terms of time and cost, and enhanced portability (Table S6). It requires only a single set of PCR reagents and liquid transfers, compared to four sets for qPCR when detecting four pathogens, reducing processing time from 8 h to just 2 h. Unlike qPCR, which solely quantifies DNA, mPCR + Nanopore provides sequence data, enabling pathogen characterization within 3 h. In contrast, qPCR requires additional Sanger sequencing, which extends the process to approximately 7 days. The system is also highly portable,

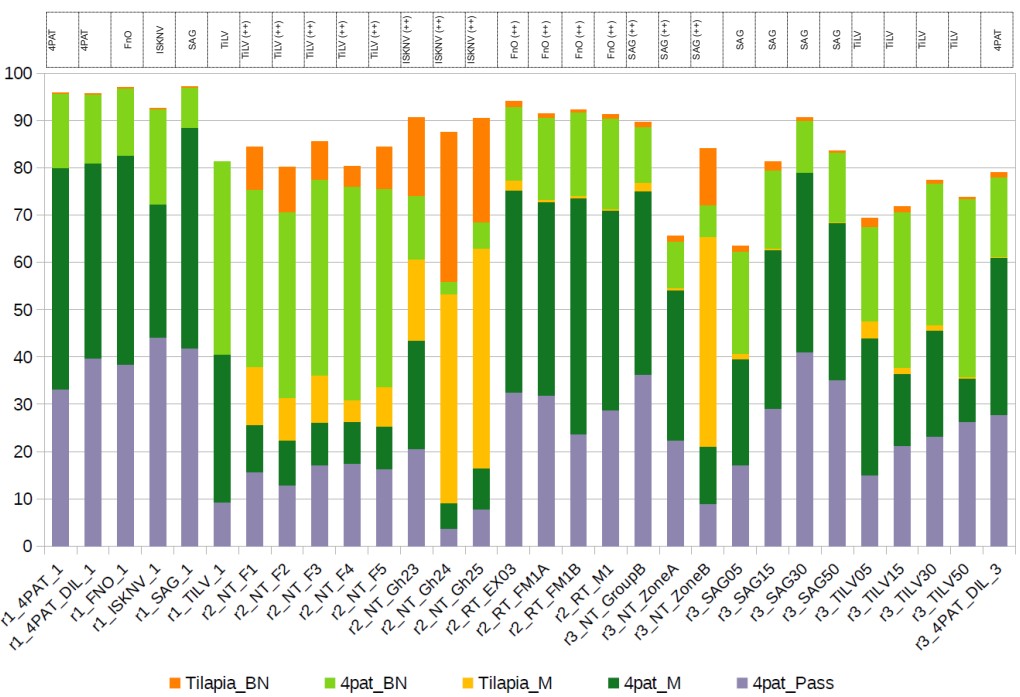

**Figure 4** **Distribution of reads assigned to the host genome and four pathogens at various stringency levels, using five different mapping strategies.** (1) 4pat_Pass, Reads that were directly assigned/mapped to one of the four pathogens using the reported stringent Minimap2 alignment strategy; (2) 4pat_M, Raw reads that failed to map to the pathogen genes using the stringent Minimap2 alignment strategy but subsequently aligned to the pathogen genes after using the default Minimap2 alignment setting; (3) Tilapia_M, Raw reads that failed to map to the pathogen genes using the stringent Minimap2 alignment approach but were subsequently aligned to the host genome after using the default Minimap2 alignment setting; (4) Tilapia_BN, Raw reads that could only be aligned to the host genome using BlastN after failing to be aligned with the default Minimap2 alignment strategy; (5) 4pat_BN, Raw reads that could only be aligned to the pathogen genes using BlastN after failing to be aligned with the default Minimap2 alignment strategy. The table above the histogram indicates previous diagnoses conducted using singleplex PCR. TiLV, tilapia lake virus; ISKNV, infectious spleen and kidney necrosis virus; FnO, *Francisella noatunensis* subsp. *orientalis*; SAG, *Streptococcus agalactiae*; ++, high pathogen load; 4PAT: mixture of PCR amplicons 4 pathogens (TiLV, ISKNV, FnO, SAG); "r1" to "r3" indicates the run number in Nanopore.

weighing only 0.1 kg, compared to the bulky 10–20 kg benchtop qPCR machine, making it significantly more suitable for field applications. Cost-wise, Nanopore sequencing is far more economical, with a per-sample cost of $6.25 compared to $40 for qPCR. Additionally, the initial system acquisition cost is substantially lower ($1,000 *vs.* $10,000–25,000 for a 96-well qPCR system). However, qPCR offers superior detection sensitivity, capable of detecting as few as 1–100 copies per reaction, whereas mPCR + Nanopore typically requires 1,000–10,000 copies per reaction, which may be crucial when detecting low pathogen loads.

## DISCUSSION

The intensification of tilapia farming has led to a surge in emerging and re-emerging infectious diseases, resulting in substantial losses in the aquaculture industry (*Machimbirike et al., 2019*; *Haenen et al., 2023*; *Shinn et al., 2023*). In regions with high-density tilapia

farming, diseases continue to evolve and intensify under strong selective pressures driven by climate change events, such as altered precipitation patterns, floodings, temperature fluctuations, and hypoxia (*De Silva & Soto, 2009*; *Subasinghe et al., 2019*). Inappropriate farming practices, such as the misuse of antibiotics, further exacerbate these challenges by contributing to the development of antimicrobial resistance (*Kathleen et al., 2016*; *Santos & Ramos, 2018*; *Pepi & Focardi, 2021*). Climate change also poses significant threats to food safety by altering pathogen growth rates and increasing the prevalence of parasites, bacteria, and viruses (*Marcogliese, 2008*; *Short, Caminade & Thomas, 2017*). Moreover, it amplifies risks to aquatic animal health by influencing pathogen virulence and host susceptibility (*Barange et al., 2018*). To address these climate-related risks, the implementation of effective, prevention-focused biosecurity plans is critical for sustainable aquaculture (*FAO, 2018*).

In this paper, we built upon our previous study (*Delamare-Deboutteville et al., 2021*), and improved the detection and genotyping of multiple pathogens by switching to multiplex PCR coupled with Nanopore sequencing. Our updated approach enabled the simultaneous and sequence-based detection and verification of four major tilapia pathogens: ISKNV, TiLV, *S. agalactiae*, and *F. orientalis*.

The need for simultaneous diagnosis of multiple pathogens is especially critical in addressing the high mortalities observed in larger grow-out tilapia. These mortalities are often exacerbated by co-infections, as highlighted by *Ramírez-Paredes et al. (2021)*, whose research demonstrated that ISKNV-positive fish frequently harbor active co-infections with *Streptococcus agalactiae* and other bacterial pathogens. This underscores the importance of comprehensive diagnostic approaches to effectively manage and mitigate such complex disease scenarios.

Our primary goal in this study is to provide sequence-based confirmation and detect pathogen variants (some may potentially be more pathogenic, warranting different management strategies), rather than focusing on quantification. Sequencing directly from PCR products allows us to confirm the presence of target sequences and identify variants, even when using relatively universal primers, without the concern of false positives common with gel-based detection methods. While high Ct values offer increased sensitivity, they limit quantification due to primer concentration and depletion, reflecting the reaction endpoint rather than the original abundance. Accurate quantification is still best achieved through quantitative PCR (qPCR); however, the equipment is costly and rarely found in resource-limited lab settings. Furthermore, sequencing-based methods are compositional rather than absolute, even with DNA spike-ins, as they measure relative proportions. Our approach offers a complementary solution, providing reliable sequence-based confirmation and variant detection, filling a useful niche alongside traditional PCR methods.

Building on the complementary nature of sequencing-based methods, our multiplex PCR-Nanopore sequencing approach demonstrated greater sensitivity, detecting pathogen sequences in samples that tested negative by qPCR. However, these findings require cautious interpretation due to the limited sample size, necessitating further validation with larger cohorts. This method offers notable efficiency advantages, enabling simultaneous detection of multiple samples and targets of varying sizes in a single reaction, thereby reducing time
and labor. Real-time sequencing with Nanopore technology provides immediate access to results as data is generated, unlike qPCR. DNA spike-ins at known concentrations can serve as positive controls, allowing for the estimation of relative abundance in positive samples and enabling operators to monitor and terminate sequencing runs once sufficient data is obtained. While the cost of sequencing reagents remains higher than that of qPCR, optimizing the number of samples and amplicon targets per library—through careful primer design while maintaining assay specificity and efficiency—can substantially reduce per-sample costs, making this approach increasingly competitive for high-throughput applications.

The discrepancies between gel-based interpretation of multiplex PCR products and qPCR or direct amplicon sequencing approaches are not surprising, given the subjective nature of gel visualization and the labor-intensive process of running multiple samples. The accuracy of pathogen detection can be significantly impacted by faint or similar-sized bands, making it challenging to standardize results. However, gel electrophoresis remains useful for validating Nanopore sequencing results. With high-degree sample-based multiplexing, it is now possible to perform high throughput sequence-based detection of multiple tilapia pathogens using Nanopore sequencing. The utilization of this application is paramount for the aquaculture industry, given that disease outbreaks are frequently linked with multiple infections (*Abdel-Latif & Khafaga, 2020*; *Huang et al., 2020*; *Liu et al., 2020*; *Basri et al., 2020*). Furthermore, as aquaculture farms are often situated in remote locations far from diagnostic reference laboratories, the ability to mobilize diagnostic testing with rapid results, high accuracy, and less complicated equipment near the farm is preferred.

Although our approach is scalable, there are limitations due to the old version of the operating system MinKNOW used during the study. Short amplicons are sequenced less efficiently and have lower read accuracy. We addressed this issue using a stringent read filtering and alignment approach, which led to a reduced read recovery in the final abundance calculation. The higher error rate of Nanopore reads has resulted in inconsistent outcomes when using different mapping alignment strategies, with more stringent parameters improving specificity at the cost of sensitivity, and *vice versa*. However, with recent advancements in Nanopore read accuracy, we anticipate this issue will diminish moving forward. These improvements are expected to enable both high sensitivity and high specificity with greater confidence, facilitating the reliable detection of pathogen and their novel variants. Some synthetic samples analyzed contained reads from other pathogens, which was unexpected and suggests cross-ligation of native barcodes during library preparation. The availability of unbound adapters is higher among samples with low DNA input, leading to higher crosstalk levels and reads from other samples. New improvements in the current Nanopore technology and library preparation protocols (as of 23 January 2025) could mitigate these issues. This includes (1) Switching to a PCR-barcoding kit to only enrich for amplicon with Nanopore partial adapter, (2) the use of a new Q20+ LSK114 kit with improved read accuracy, and (3) the use of high accuracy Dorado base-calling model (https://github.com/nanoporetech/dorado).

By enforcing a stringent alignment setting, the reads aligning to the pathogen gene segment are directly suitable for subsequent reference-based variant calling, producing

highly accurate variants useful for biological interpretation. This is exemplified by the consistent recovery of a TiLV segment 9 variant previously only found in a Vietnamese TiLV strain in all Thai tilapia samples from the same sampling batch. Considering the proximity of Thailand and Vietnam, this may suggest a possible transboundary event, highlighting the need for more thorough sampling and sequencing of additional segments to elucidate the degree of TiLV diversity in both regions.

For field diagnostics, some of the equipment used in the present study can be replaced with portable items that are small enough to fit in a bag for use in remote settings. An example of the field application of this diagnostic workflow is the Lab-in-a-backpack concept developed by WorldFish (*WorldFish, 2020*; *Huso et al., 2020*; *Cagua et al., 2021*; *The University of Queensland, 2021*; *Chadag et al., 2021*; *Barnes et al., 2021*). It includes all the necessary sampling, extraction, and sequencing kits, pipettes, and consumables, as well as the use of the miniPCR thermocycler and blueGel electrophoretic system from miniPCR bio™, a minicentrifuge, a magnetic rack, a fluorometer, a Nanopore MinION sequencer, and flow cells for sequencing. Additionally, a laptop with minimum requirements to run the MinKNOW software is required. Similar PCR-sequencing approaches using portable equipment have been employed for the rapid identification of a diverse range of biological specimens, including plants, insects, reptiles (*Pomerantz et al., 2022*), and viruses such as COVID-19, Ebola, and Zika (*González-González et al., 2019*; *González-González et al., 2020*).

Improvements in the remote and low-resource applications of the current methodology will broaden its use for capacity building and implementations in low-income countries, where high mortalities due to unknown causes are prevalent in tilapia farming. Accurate and rapid genomic detection of fish pathogens through our amplicon-based approach will allow health professionals to promptly provide advice and take necessary actions for producers. Moreover, it can facilitate informed decisions regarding additional investigations, such as whole-genome sequencing (WGS), on isolates stored in biobanks. The sequence data obtained from WGS offer crucial epidemiological insights, which can be utilized to develop customized multivalent autogenous vaccines using local pathogens (*Barnes et al., 2022*). These vaccines can then be administered to broodstock and seeds before distribution among grow-out farmers for restocking purposes.

## CONCLUSIONS

In summary, our study presents an attractive approach for detecting and verifying four tilapia pathogens in clinically sick fish, which can also be applied to pathogens in other livestock, fish, or crustaceans if the genetic information of the pathogen is publicly available for primer design and reference mapping. Although there are limitations to the current pipeline version used at the time of this study, we are optimistic that the current improvements in Nanopore technology will further enhance the accuracy and scalability of our approach.

## ACKNOWLEDGEMENTS

We want to extend our utmost gratitude and profound recognition to the late Dr. Pattanapon Kayansamruaj for his valuable contributions during the early stages of conceiving this research endeavor.

### Funding

This work received financial support from Norway (Project Title: Increased Sustainability in the Aquaculture Sector in Sub-Saharan Africa, through Improved Aquatic Animal Health Management), the CGIAR Initiatives on Aquatic Foods and Protecting Human Health Through a One Health Approach, and the CGIAR research program on Sustainable Animal and Aquatic Food. The contents of this publication are the sole responsibility of the authors and can in no way be taken to reflect the views of the donors and the Government of Norway. The funders had no role in study design, data collection and analysis, decision to publish, or preparation of the manuscript.

### Grant Disclosures

The following grant information was disclosed by the authors:
Increased Sustainability in the Aquaculture Sector in Sub-Saharan Africa.
CGIAR Initiatives on Aquatic Foods.
Government of Norway.

### Competing Interests

The authors declare there are no competing interests. Jérôme Delamare-Deboutteville, Laura Khor Li Imm and Chadag Vishnumurthy Mohan are employed by WorldFish, and Han Ming Gan is employed by Patriot Biotech Sdn Bhd.

### Author Contributions

- Jérôme Delamare-Deboutteville conceived and designed the experiments, performed the experiments, analyzed the data, prepared figures and/or tables, authored or reviewed drafts of the article, and approved the final draft.
- Watcharachai Meemetta performed the experiments, prepared figures and/or tables, and approved the final draft.
- Khaettareeya Pimsannil performed the experiments, prepared figures and/or tables, and approved the final draft.
- Han Ming Gan analyzed the data, prepared figures and/or tables, authored or reviewed drafts of the article, and approved the final draft.
- Laura Khor analyzed the data, authored or reviewed drafts of the article, and approved the final draft.
- Mohan Chadag conceived and designed the experiments, analyzed the data, authored or reviewed drafts of the article, and approved the final draft.

- Ha Thanh Dong conceived and designed the experiments, analyzed the data, prepared figures and/or tables, authored or reviewed drafts of the article, and approved the final draft.
- Saengchan Senapin conceived and designed the experiments, analyzed the data, prepared figures and/or tables, authored or reviewed drafts of the article, and approved the final draft.

## DNA Deposition

The following information was supplied regarding the deposition of DNA sequences:

The demultiplexed FastQ files for all 30 samples are available at BioProject: PRJNA957495; SAMN34371653–SAMN34371700.

## Data Availability

The Linux scripts used to generate initial FastQ files, assembled amplicons (public and from this study) are publicly available at Zenodo: Jerome Delamare-Deboutteville, Han Ming Gan, Meemetta Watcharachai, Ha Thanh Dong, Chadag Vishnumurthy Mohan, Saengchan Senapinx, & Laura Khor Li Imm. (2023). Dataset for Multiplex-PCR detection and Nanopore-based genotyping of fish pathogens (v2-260423) [Data set]. Zenodo. https://doi.org/10.5281/zenodo.7866295.

The intermediary files generated during the bioinformatic analyses are publicly available at Zenodo: Jerome Delamare-Deboutteville, Han Ming Gan, Meemetta Watcharachai, Ha Thanh Dong, Chadag Vishnumurthy Mohan, Saengchan Senapinx, & Laura Khor Li Imm. (2023). Dataset for Multiplex-PCR detection and Nanopore-based genotyping of fish pathogens (v2-260423) [Data set]. Zenodo. https://doi.org/10.5281/zenodo.7866295.

## Supplemental Information

Supplemental information for this article can be found online at http://dx.doi.org/10.7717/peerj.19425#supplemental-information.

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
