# Peer review of "Multiplex polymerase chain reaction (PCR) with Nanopore sequencing for sequence-based detection of four tilapia pathogens"

_PeerJ, doi:10.7717/peerj.19425_

## Round 0.1 · original submission · Major Revisions

The manuscript entitled "Multiplex PCR with Nanopore sequencing for sequence-based detection of four tilapia pathogens" proposes a diagnostic approach combining PCR amplification with nanopore sequencing for the identification of key tilapia pathogens. While the study presents a method with potential practical applications, it requires substantial revisions to clarify the rationale, significance, and novel contributions of this approach.

Major Points for Revision:

Rationale and Significance of the Methodology:

The introduction should better explain the practical advantages of this method over traditional qPCR. Specifically, it would benefit from addressing whether the proposed approach offers measurable improvements in sensitivity, specificity, cost-efficiency, or time savings. Currently, it is unclear if this approach is necessary when conventional PCR or qPCR may be sufficient for pathogen detection.
Additionally, authors should discuss why nanopore sequencing is advantageous for this application, particularly given the short (~200 bp) amplicons generated by PCR. Clarifying the added value of sequencing for precise identification or strain-level differentiation would strengthen the justification for this method.
Methodological Details and Validation:

The authors should provide more background on multiplex PCR, discussing its strengths and limitations relative to sequencing. Given that sequencing verification is crucial for pathogen confirmation, the manuscript would benefit from reorganizing the explanation to first introduce the strengths of multiplex PCR, followed by the limitations of Sanger sequencing and the relevance of nanopore sequencing for accurate pathogen detection in real-world scenarios.

There is a need to address differences between qPCR sensitivity and the proposed method, possibly with a comparative study to illustrate if the new method is equally or more sensitive.

Line 299 suggests eliminating gel electrophoresis in favor of nanopore sequencing. It is recommended that instead of omitting gel electrophoresis, it should be validated with nanopore sequencing results to ensure reliability.
Supplementary materials, particularly Figure 1 and Table 5, require more clarity. Figure 1 should contain higher quality images of gel electrophoresis results, and Table 5 would benefit from additional contextual information, such as aligning columns with Table 1 for consistency.

Pathogen Impact and Co-infection Detection:

The manuscript should expand on the significance of the four pathogens (TiLV, ISKNV, Francisella orientalis, and Streptococcus agalactiae), particularly their impact on farmed tilapia in specific regions like Southeast Asia. This background would help emphasize the broader relevance of the study.

Given that the method has potential value for detecting co-infections, a dedicated section in the Results is recommended. The authors should clarify whether co-infections were detected after nanopore sequencing, potentially integrating PCR and sequencing data for a more comprehensive analysis.
Discussion and Quantifiable Benefits:

The Discussion should delve deeper into the quantifiable benefits of this method compared to qPCR, covering factors such as time, cost, and diagnostic accuracy. For example, specific details on how long this method takes relative to qPCR would provide a clearer perspective on its practical utility.

Since pathogen evolution rather than the disease itself is relevant (lines 283-284), it would be useful to include examples of selective pressures that drive pathogen evolution, supported by relevant references.

It would also be beneficial to discuss the potential of this method for quantitative analysis, not just detection, and to outline how differences in sequencing read numbers might reflect pathogen abundance. This could involve using a DNA spike-in series at different concentrations as a control to validate quantification potential.

Technical Clarifications for Reproducibility:

To improve reproducibility, the bioinformatics scripts should be thoroughly commented, and a workflow diagram for analysis steps should be included in the paper. Additionally, specifying software versions and adding example commands would enhance clarity for readers seeking to replicate the analysis.

In conclusion, this manuscript has promising elements but requires a clearer demonstration of the novel aspects, advantages, and practical relevance of this combined PCR-nanopore approach. By addressing the above points, the manuscript would better communicate its value to the research community.

Thus, please reply to each concern of this Editor and the detailed analysis of the reviewers.

Reviewer 1 ·

Basic reporting

1. Lines 55-8: update to most recent FAO report (2024).
2. Lines 63-5: the authors are encouraged to provide more details on the impact of these four pathogens. Are they only important in farmed tilapia in Southeast Asia?
3. Lines 76-80. Consider restructuring the paragraph. First explain what multiplex PCR is, its strengths and why sequence verification is required. Then discuss the limitations of Sanger sequencing and why ONT is more appropriate for real-world situations (i.e. why you have chosen it).
4. 283-284: It is the pathogen that evolves, not the disease. Provide examples of selective pressures with references.
5. The paragraph on 290 is vague.
6. Line 299: Consider removing. I don’t think that gel electrophoresis should be eliminated - it should be validated through Nanopore sequencing.
7. Necessary to discuss the differences in results among mapping parameters.
8. Line 324: If these were the same fish species, consider changing the word “spillover” as this reflects cross-species transmission. Important to mention that this comparison is based on a fragment of segment 9 and that more sequencing is required.

Experimental design

No comment.

Validity of the findings

Since this pipeline is valuable for detecting co-infections, it would benefit from a small dedicated section in the Results, as it is currently unclear whether the authors identified any co-infections. This could combine PCR with sequencing data. For example in lines 223-224 did you detect these co-infections after nanopore sequencing?

Reviewer 2 ·

Basic reporting

The manuscript titled "Multiplex PCR with Nanopore sequencing for sequence-based detection of four tilapia pathogens" described an approach for tilapia lake virus (TiLV), infectious spleen and kidney necrosis virus (ISKNV), Francisella orientalis, and Streptococcus agalactiae. Authors proposed using PCR amplification together with nanopore sequencing for diagnostics of major tilapia pathogens.

The main concern about this paper is the overall significance of the proposed approach because of few points: 1) The pathogens can be easily detected and distinguished by conventional PCR assay, as authors also described in the MS, and it seems that nanopore sequencing is useless with these primer combinations; 2) Authors used long-read sequencing to sequence quite short PCR products (near 200bp); 3) Author did not provide any advantages of application of nanopore sequencing compared to simple PCR with the same primers, that can include, for example, precise identification races of the pathogens that can not be achieved by simple PCR.

Experimental design

no comment

Validity of the findings

I do not feel that the described results provide any kind of significant novelties and advantages.

Additional comments

no comment

Reviewer 3 ·

Basic reporting

The paper provides a novel diagnostic method to detect 4 tilapia pathogen. The study uses both multiplex PCR along with nanopore to validate the results. The study has very practical uses for use at farm sites. The approach using nanopore is especially useful to not only to as diagnostic, but as additional validation.

Experimental design

Please discuss why the l0^4 copies/reaction for Francisella orientalis compared to the other pathogens

Validity of the findings

Code:
the scripts needs to be commented and documented. A workflow diagram in the paper will be really helpful.
- Please also note the versions used.
- Please note how to run the analysis scripts and any examples

Supplemental Figure 1: Please describe the lanes

Supplemental Table 5: Please revise this table. It is hard to understand the sample being referred to here. Please add information from Table 1 as a column to make this table readable

Gel electrophoresis images are not clear. Higher quality images are needed.

Please discuss correlation between the gel bands and sequencing results

Annotated reviews are not available for download in order to protect the identity of reviewers who chose to remain anonymous.

Reviewer 4 ·

Basic reporting

The study by itself is not complex and easy to follow. However as a methodology work, it will be better if the authors could better elaborate the rationale to develop this methodology, in the intro section. For example, compared with traditional qPCR methodology, what is the value of this new method? Does it save money and time, and to what degree? How much convenience have it really brought? On top of convenience, what are the new info revealed and why is that info important? Current version does not have enough discussion at the intro section and throughout the work, therefore I don’t fully recognize the value from the methodology (other than validating that it works).

Experimental design

I think the work has several major issues:
• Like mentioned above, because the authors haven’t elaborated the value of the method well, I don’t quite get the significance of this work. From a first look, I feel like this method is just a combination of different PCR reactions into one reaction and then followed by nanopore sequencing. If qPCR can already detect pathogen in a very sensitive manner (though needing several different qPCR reactions – but I really don’t see that as a barrier), why is it necessary to do it this way?
• Line 211-214: Would like to see a comparison of sensitivity of this method vs. qPCR (currently only mention this method’s sensitivity). If not sensitive enough, the value of the method will be greatly challenged.
• Line 236-237: The author mentioned this method could detect variant – which I agree that is an extra info that can’t be provided by qPCR. However why is the variant info important? The authors needs to better elaborate the rationale (rather than just stating the fact that variants are detected).
• Line 250-255: Figure 3 shows a comparison between qPCR and this method. However the 2 method, while directionally consistent, still have different patterns. Which one is the gold standard (I assume qPCR)? If qPCR is the gold standard, does the inconsistency makes the nanopore-methodology less valuable?
• Line 256-262: Could the sequencing read number imply the abundance of pathogen in the first place (or not)? I prefer to see some discussion on whether this method could serve quantification purpose (not just detection). In addition, if the authors would like to use this method for quantification purpose, they need a quantifiable control series (e.g. DNA spike-ins at different concentrations).
• Discussion: I would like to see more quantifiable benefit. For example, how long does qPCR take vs. this method? How do the costs of traditional and new method compared?

Validity of the findings

Again as mentioned, I see the major issue being “why do we need this method” and “is it novel enough to publish a paper“, rather than this method won’t work. Overall the study is a bit too simple, and not yet demonstrating its novelty or significance in the current format.

---

## Round 0.2 · Minor Revisions

Dear Dr. Delamare-Deboutteville,

Based on the concerns raised, I would recommend a Minor Revision decision.

While the manuscript presents an innovative approach, it requires additional data and clarifications to strengthen its validity and practical applicability. The key points that need to be addressed include (as stated in the previous decision):

Quantitative Comparisons with qPCR

Provide specific data on sensitivity, specificity, cost, and time-to-result relative to qPCR. Include a clear comparison table if possible.
Justification of Nanopore Over qPCR

Address the claim that qPCR equipment is a barrier, providing supporting data or context.
Justify why the method is superior beyond its multiplexing capability.
Scalability and Broader Applicability

Discuss whether the method can be scaled for more than four pathogens, which would make its application more compelling.
Clarification of Detection Limits

Clearly compare the detection limits of the proposed method versus qPCR to contextualize its diagnostic utility.
Decision: Minor Revision
The study is well-structured and has potential, but it requires further clarification and minor adjustments to enhance its impact and justify its significance in pathogen detection for aquaculture.

Reviewer 1 ·

Basic reporting

Overall, I believe that the authors have adequately addressed my comments, and that the additional paragraphs strengthen the manuscript.

Experimental design

No comment

Validity of the findings

Again, I believe that the authors have adequately addressed my comments.

Reviewer 4 ·

Basic reporting

No comments.

Experimental design

Same as above, I'm not fully convinced that the method proposed in this paper is of substantial significance for publishing on Peer J (especially when it is compared with qPCR, the gold standard). The authors have added some discussions, unfortunately no major changes in terms of study per se.

Compared with qPCR, I'm not convinced by the argument that qPCR equipment is a barrier (to be honest this is an equipment that is more commonly used by labs today vs. Nanopore). I'm also not very convinced that it is a troublesome process to conduct single-plex qPCR just for 4 pathogens (this argument only works if we are looking at a lot more pathogens simultaneously, e.g. 20+ - however the study per se has not shown whether the method could be used to detected a large number of common pathogens beyond just 4, which will be a more interesting application).

Validity of the findings

While I suggested the authors to provide more comparison over qPCR in terms of time, sensitivity, cost, there are no quantitative comparison (though the authors have added qualitative statements suggesting that sensitivity of the method is lower while cost is higher). This directional assessment has made readers hard to conclude how large disadvantages these will be.

---

## Round 0.3 · accepted · Accept

Congratulations on the acceptance of your manuscript.